# Resettlement, Employment, and Mental Health Among Syrian Refugee Men in Canada: An Intersectional Study Using Photovoice

**DOI:** 10.3390/ijerph21121600

**Published:** 2024-11-30

**Authors:** Nancy Clark, Gökce Yurdakul, Carla Hilario, Heba Elgharbawy, Nedal Izzden, Elias Moses, Muna Zaidalkilani

**Affiliations:** 1School of Nursing, Faculty of Human and Social Development, University of Victoria, Victoria, BC V8W 2Y2, Canada; 2Department of Diversity and Social Conflict, Institute of Social Sciences, Humboldt-Universität zu Berlin, 10117 Berlin, Germany; gokce.yurdakul@sowi.hu-berlin.de; 3School of Nursing, Faculty of Health and Social Development, The University of British Columbia, Okanagan Campus, Kelowna, BC V1V 1V7, Canada; carla.hilario@ubc.ca; 4Department of Psychology, University of Victoria, Victoria, BC V8P 5C2, Canada; hebael@uvic.ca; 5University of Victoria, Victoria, BC V8W 2Y2, Canada; nedalizdden@gmail.com (N.I.); elias.moses@uvic.ca (E.M.); 6Counselling Psychology, Yorkville University, Toronto, ON M4Y 1W9, Canada; muna.zaidalkilani@yorkvilleu.ca

**Keywords:** men, mental health, social determinants, masculinities, intersectionality, photovoice, community-based participatory action research, forced migration, Syrian refugee, employment, economic integration

## Abstract

Context: The impact of forced migration on the mental health of refugee men is far-reaching and compounded by gendered masculinity, which shapes men’s access to employment and other resources. A gap in knowledge exists on the broader determinants of refugee men’s mental health. Methodology: Using community-based participatory action research and the arts-based method of photovoice, this study advances knowledge about the gendered impacts of forced migration from the perspective of (*n* = 11) Syrian refugee men in the Canadian context. Theoretical approaches of intersectionality and masculinity were applied to understand how power relations shape Syrian men’s identities, their access to employment, and impacts on their mental health. Analysis and Results: Syrian men’s identities were marginalized by working in low-wage jobs because of inequitable policies that favored Canadian experience and credentialing assessment processes that devalued their knowledge. Multiple and overlapping factors shaped Syrian men’s mental health including language and literacy barriers, time and stage of life, isolation and loneliness, belonging and identity, and gender-based stress. Caring masculinities performed through fathering, cultural connection, and service-based work promoted agency, hope, and resilience. Conclusions: Public health and community-based pathways must adopt gender-responsive and intersectional approaches to policy and practice. Peer-based programs may mitigate harmful forms of masculinity and promote transformative change to support refugee men’s mental health.

## 1. Introduction

International migration trends related to forced displacement have showed settlement challenges along the lines of gender inequities. The World Migration Report (2024) describes a global trend in the masculinization of migration, which reflects an increase in male labor migration, but also a lack of a gender-responsive and nuanced understanding of how gender operates across geographies, labor integration, violence, and trauma [1]. In this context, there exists a gender bias which constructs the migrant man as the breadwinner and decision maker, reinforcing hegemonic masculinity. Refugee men are a subset of migrant men who may experience hegemonic masculinity through complex expressions of power and privilege and across other factors related to their abilities, race/ethnicity, and class [2,3]. Hegemonic masculinity—that is, masculinity which threatens models of male gender hierarchies of power through authority and strength—may be the most harmful form of masculinity [4,5,6,7]. Importantly, dominant forms of masculinity are maintained through structural processes which maintain power, privilege, and dominance over subgroups of men, women, and other genders [2,3]. While many migrant women participate in caregiving and unpaid work and experience vulnerability, scholars have argued that gendered stereotypes have masked the nuanced vulnerabilities of refugee men, perpetuating the idea that male refugees are a threat to their countries of settlement, and reinforcing refugee women as victims and in need of care [1,8,9,10,11,12,13,14]. These stereotypes may bias migration policies and practices which promote gender-based analysis touted by global and national migration policies [1,15].

Masculinities include many practices of men’s gender roles, relations, and identities and are emerging as an important theoretical orientation toward understanding the health equity and mental health of men [4,16]. However, research is lacking on the intersections of migration, gender, and men’s mental health, alongside other determinants including, but not limited to, race/ethnicity, discrimination, age, sexuality, and class [2,16]. The case of the Syrian refugee crisis is one of the world’s largest causes of forced displacement due to war and violence, with over 5.3 million Syrian refugees in Türkiye, Lebanon, Jordan, Iraq, Egypt, and North Africa [17]. In both pre- and post-displacement contexts, Syrian men may experience different forms of masculinity, including disparate treatment and disadvantage based on race/ethnicity, class status, downgrading of skills, and limited or precarious work. The refugee crisis has also created anti-immigration rhetoric, specifically in settlement countries’ mainstream media and politics, as Syrian men are stereotyped as ‘bogus’ refugees and perpetrators of violence and of Arab patriarchal norms [14,18]. Marginalized masculinities are often experienced by racialized working-class men locked out of power and care [7,19,20]. In this context, discrimination, stigma, and hegemonic masculinity make it particularly challenging to elicit refugee men’s vulnerabilities and stressors associated with migration, economic integration, and affects on men’s mental health.

Studies suggest that Syrian men displaced by war may experience increased vulnerability to mental health problems related to barriers to employment and exploitation affecting family health [21,22,23,24,25,26]. A study conducted in Jordan, surveying 534 households, showed that Syrian men expressed feelings of depression and generalized anxiety over the safety of their families and concerns about poor working conditions and exploitation in low-paying jobs [10]. Political and social factors related to labor exploitation, long-distance work, time away from family, and mandatory military services also affect Syrian men and family health, contributing to disruption of gender roles and cultural norms and increased potential for gender-based violence [10,12,22]. The root causes of gender-based violence can be related to hegemonic masculinity and other structural processes that reinforce gender inequalities. In the Canadian context, a gap in knowledge exists on the factors that shape Syrian men’s mental health. However, studies show that Syrians who resettled through government assistance programs (GARS) experience the lowest pay, precarious employment, and insecure or unstable housing [26,27,28,29,30,31]. In addition, most Syrians are employed in sectors that typically provide entry-level opportunities, and only one in five reported being in a job or occupation that matched their education, skills, and experience [27]. Lack of credential recognition of skills further marginalizes migrant men and poses mental health challenges, whose identities are strongly tied to their job and occupations [32,33]. Because of the challenges of economic integration, many migrants engage in non-standard employment known as ‘platform’ or ‘gig’ work which is gendered differently across geographies [34,35,36,37]. Through these platforms, work is performed offline (e.g., ride-hailing, food delivery services); “platformed distinction” work is often viewed as a low barrier to labor market integration and may enhance economic integration in the short term, but also creates positions of marginalization and economic downgrading that are specific to immigrants [36,37]. During forced migration and resettlement, refugee men need to continually adjust to changes in their gender roles and expectations, particularly for families in which fathers are not the main contributor to household income [12,38]. Syrian refugee men ground their identity in their roles as providers and fathers and find ways to reconstitute their power through caring practices [39,40,41]. Service-based work and fathering problematize normative masculine ideals and can be conceptualized through ‘caring masculinities’. Caring masculinity stem from the margins of society and considers working-class men as productive agents of change and embraces values of care such as positive emotion, interdependence, and relationality [19,20].

Despite processes which embrace values of care, refugee men may have less access to resources and less dedicated support for their mental health, contributing to lower help-seeking patterns [41,42,43]. Research on men’s mental health and employment has shown that both life transitions and parenthood are predictive factors for depression among men [32,33]. Moreover, chronic and acute stressors related to unemployment can lead to severe negative mental health consequences, as norms of masculinity may contribute to men’s suicidal ideation, particularly for men who strongly identify as ‘self-reliant’ [16,33,43]. When men do seek out help, it is more likely to be from peers, but there is often a lack of formalized peer support networks for men and boys [8,16,43,44,45]. Studies on men and mental health suggest that men struggle with mental health issues, including trauma, and are less likely than women to access mental health services due to stigma [8,44]. Stigma maybe a key for help-seeking for men in Arab social and cultural contexts [10,25,39]. Barriers in access to mental health services and support for migrant, racialized men may be related to broader processes of normative masculinity, in which strength is often equated with stoicism and passed down from one generation to the next [44].

The implication of gender is far-reaching and is often not well integrated with public health policies and practices. In addition, gendered impacts of migration must be viewed longitudinally as many of the barriers to economic and social capital are compounded during resettlement [21,45,46]. A gap in research exists on wider social determinants that shape men’s mental health, including perspectives from men marginalized by their race/ethnicity, refugee status, employment, and other social categories [2,43]. The purpose of this research is to understand Syrian men’s perspectives on the factors that shape their mental health and wellbeing in the context of their resettlement in Canada, with a specific focus on the effects of masculinity on their resettlement and employment experiences. The research questions guiding this study are as follows:What do Syrian men perceive as important factors in shaping their mental health and wellbeing in the context of their participation in employment?How does unemployment or underemployment impact Syrian refugee men’s mental health and wellbeing?What are Syrian refugee men’s gendered experiences of economic integration?

## 2. Materials and Methods

A qualitative research design guided by reflexive thematic analysis (TA) and [47,48,49] principles of community-based participatory action research (CBPAR) were used to understand Syrian men’s perspectives and experiences of resettlement, economic integration, and the factors which shaped their mental health [50,51,52]. A qualitative reflexive thematic analysis (TA) embraces nuance and complexity where knowledge comes from active engagement with the data; interpretation is subjective and interpretive [48,49]. In addition, reflexive TA is flexible in adapting critical theoretical approaches to qualitative research [48]. Both theories of masculinity and intersectionality were used to capture the multiple dimensions that structure Syrian men’s mental health, including their experience of employment and other axes of inequity. In this way, knowledge is situated and grounded in the lived realities of Syrian men. CBPAR provided a framework for allowing community members, researchers, and Syrian men to collectively inform the research process and outcomes, thus leveraging the knowledge of diverse community members for the purpose of positive social change [51,52]. In this approach, we aimed to build equitable relationships with Syrian men and trust by promoting community strengths. The impetus for the study came from an identified need in the settlement service sector to understand challenges Syrian men were experiencing during their resettlement. The broader community, i.e., settlement service providers, employment counselors, and peer researchers, provided input about data collection, recruitment, and dissemination of research findings through a community advisory board [51].

Photovoice is a participatory arts-based method, used to not only tell stories through photographs but to democratize how knowledge will be used, and is increasingly being used in research with people who experience forced displacement, as well as with those that are marginalized and dispossessed [53,54,55,56,57,58,59]. Fitting with qualitative reflexive TA, photovoice workshops occurred on weekends at the settlement service agency because it was familiar and easily accessible for the research participants. Participation was voluntary and ongoing throughout the study. Settlement service providers were not informed about which men participated. In this study, photo elicitation interviews were also conducted through Zoom technology to facilitate a deeper understanding about Syrian men’s experiences and make meaning of the photos and provided initial analysis to guide the photovoice workshops with Syrian men. Online photovoice methods offer flexibility for research participants [58]. This study occurred between 10 May 2022 and 9 May 2024, and included three phases:Phase I—partnership building and community participation;Phase II—data collection and analysis;Phase III—knowledge mobilization.

### 2.1. Phase I—Partnership Building and Community Participation

In this phase, we developed a community advisory board (CAB) [60,61], recruited participants, and provided an initial photovoice workshop. Syrian men were invited to participate in a 2.5-day photovoice training workshop that provided information about the ethics of using photovoice and the use of a camera or smart phone and offered some prompts to guide their photography. Prompts for taking photographs included pictures of the men’s work, pictures that challenged or supported their mental health, and pictures that supported resilience and wellbeing. Participants were also provided with photovoice worksheets so that they could journal and make notes with their photos. Two Syrian men with refugee backgrounds were hired as peer researchers to build trust and engagement throughout the research process [59,60,61]. Peer researchers in our study assisted with translating and interpretating all materials from Arabic to English, co-developing the photovoice workshops, recruiting participants, obtaining informed consent, and co-authoring published materials. The peer researchers also participated in the study as interviewees and in the photovoice sessions. This process facilitated mutual sharing of men’s stories.

CAB members included peer researchers, research team members, counselors, employment specialists, and staff members in not-for-profit immigrant service organizations. Over the course of the research, four community advisory meetings were held: two meetings in 2021–22 and two meetings in 2022–24. Participant recruitment was facilitated by an information session at the settlement agency who facilitated recruitment through a poster and word of mouth. Syrian men were purposefully sampled, meaning that the men were purposefully recruited because they would be able to address the research questions, due to their knowledge and expertise [47]. Men were included if they arrived in Canada as a refugee or asylum seeker or through any government sponsorship arrangement, and who were between the ages of 19 and 65. A total of 15 refugee men from Syria attended the information session and were provided information about how to contact the peer researchers through WhatsApp Business on their smartphone. From the information-sharing event, nine Syrian men were recruited to participate in the study; with these recruited participants and the two peer researchers as participants, our study had a total of *n* = 11 participants. Two of the men were hired as peer researchers and also participated in the data collection [62,63].

### 2.2. Phase II—Data Collection and Analysis

This phase consisted of a photo-taking period, the photo elicitation interviews, and a second photovoice workshop. Syrian men engaged in photo-taking activities, which consisted of taking photographs on their smartphone or camera over a two-month period (August–September 2022). The men also had the option of sharing photographs that they had previously taken. Eleven photo elicitation interviews occurred between September and November 2022. All but one photo elicitation interview occurred online over Zoom; those same ten interviews were conducted in Arabic with an interpreter. The in-person interview was conducted in English. Prior to the interviews, the men uploaded their photographs onto a secure shared drive so that they could be shared easily during the online interviews for further discussion. Detailed reflexive notes were taken during the interviews to document which photographs the men wanted to discuss during photovoice workshops and their relevance to the study research questions. Interviews lasted 60–90 min and were translated verbatim into English, and then these transcripts were uploaded into the NVivo Version 12 data management software program. Translation of interviews in Arabic was validated by two Arabic- and English-speaking research assistants on the project (EM and NZ).

Using reflexive TA, open coding was used to identify text segments that related to the photographs. Syrian men’s narratives were coded, categorized, and continually refined until preliminary candidate themes [49] were developed by the study PI (NC). Candidate themes were sorted and linked to the photographs the men described as relevant to the research study questions. Intersectionality was used as an analytic tool to uncover the multi-dimensional and complex experiences of Syrian men. This process allowed for a nuanced, multi-level analysis that linked individual men’s experiences with broader structures and systems of power [64,65,66,67]. The decision to use an intersectional analysis with masculinity theory was based on critiques of hegemonic masculinity which fail to capture diverse masculinities within diverse groups of men [67]. Importantly, when intersectionality is combined with hegemonic masculinities, complex expressions of power and privilege are revealed [67].

Applying an intersectional lens meant that candidate themes could capture multiple facets of an idea or concept through one photograph. After a preliminary analytic process, the study PI created eleven individual photovoice workbooks to capture key photographs and corresponding themes unique to each of the men. The workbooks were used to further validate and co-construct the meaning of the photographs in the second photovoice workshop. Through this process, the men provided their interpretation of the photographs in either Arabic or English and identified which photographs they wanted to use for knowledge dissemination. The participants could also add concepts or ideas that were relevant or missing.

By using a reflexive thematic TA, the peer researchers and the study PI collectively reflected our individual subjectivities and experiences; this included shared experiences of coming to Canada from diverse Arabic cultural backgrounds—the peer researchers being from different regions in Syria, and the study PI having Palestinian heritage—with the 11 research participants. The PI and peer researchers also created a PowerPoint presentation to facilitate refining of the themes, their meaning, and links to the photographs. Through a reflexive TA process, Syrian men shared their collective experiences and interpretation of their photos. Study rigor was supported by use of purposeful sampling, photovoice workshops, the CAB, and quotes from the men’s interviews to support the findings.

### 2.3. Phase III—Knowledge Mobilization

Photovoice methods can produce challenges for knowledge mobilization and useful information for knowledge users, i.e., policy makers and practitioners [58]. During this phase, all 11 men participated in co-developing a digital storybook, a video, and an online article to raise awareness about Syrian men’s mental health and recommendations for employers and settlement counselors on inclusive, gender-responsive policies and practices. These knowledge outputs were shared with local and national immigrant and refugee settlement services that provide employment and counseling support services to refugee men. The co-developed video was embedded in the digital storybook, which was then shared across local settlement agencies and networks including The Centre for Addiction and Mental Health (CAMH): https://irmhpcoursestage.netlify.app/sitecore/newsletter/evidencesnapshot.html (accessed on 26 November 2024). 

#### Ethics

This study received ethics clearance from the university’s Human Research Ethics Board (#21-0538). Written and oral consent was obtained from all research participants, and consent was ongoing throughout the project. Each of the 11 men, including the peer researchers, received a CAD 50 honorarium for each hour of work, including the photo elicitation interviews, photovoice workshops, and meetings. All in-person photovoice workshops included breaks for prayers for those who identified as Muslim, culturally relevant food, child-minding support, and renumeration of transportation costs. Due to the highly stigmatized nature of mental health, participants’ privacy and anonymity have been respected by removing participants’ names and replacing these with code numbers in the reporting of the results.

## 3. Results

### 3.1. Participants

The mean age of the participating men was 39.1 and their ages ranged from 21 to 51 years. All men were in rental housing during the study, with 6 of 11 men residing in subsidized housing. All men self-identified as Syrian, and one identified as a Palestinian refugee living in Syria. Only one participant was born outside of Syria, in Kuwait. Most men had completed a high level of education—a master’s or bachelor’s degree. Nine out of eleven men arrived with nuclear families; two men were single. Their occupations and professions included student, civil engineer, dentist, musician, agriculturalist, architect, factory worker, electrician, chartered accountant, baker, and customer service provider. Eight out of eleven men were employed, and only two were employed in jobs that matched their work experience and professions. Many men described working multiple jobs as, for example, a delivery driver, Uber driver, cook, painter, and shelf stocker at a food distribution warehouse.

Most identified as Muslim; two were Christian, and two identified as having no religion or as atheist. During the study, six men had already received Canadian citizenship status and five were permanent residents. Six out of eleven men arrived under government-assisted refugee status, three were privately sponsored, and two arrived with asylum status. The average approximate household income was CAD 42,000/year, and the average family household had 3.3 people. The small family size may account for the fact that most participants had extended and other family members who had not received refugee status in Canada. All the men had good conversational English, and their average Canadian Language Benchmark was level 6 [68]. Although the Syrian men’s demographic data show a heterogenous sample, all men experienced some form of marginalization based on their lower social class status, ethnicity, and inability to find meaningful work. This suggests that Syrian men’s identities are shaped by dominant forms of hegemonic masculinity that structure their access to employment and economic integration.

All eleven men arrived in Canada between 2016 and 2022, with four arriving during the COVID-19 pandemic. The men’s demographic data can be found in Table 1.

The following themes highlight Syrian men’s perspectives on the intersecting factors shaping their mental health, and how power is structured along these intersecting categories of identity that include (1) language barriers, (2) time and stage of life, (3) isolation and loneliness, (4) belonging and identity, and (5) gender-based stress.

### 3.2. Language and Literacy Barriers: “A Person Is Protected if They Have a Strong Language”

Most men considered the English language to be their biggest barrier to accessing meaningful employment in Canada. For some of the men, English language proficiency created a sense of precarity and vulnerability during resettlement. As one man explained:

“*It’s tough to see the war, but just do not give up. Like, keep trying … because like sometime, when I came here, I opened some website to look for a job, but when I see the job description and they’re all in English and I barely understood some English, … and just give up. Oh, it’s hard to find a job if I can’t understand the job description. … I give up a lot*”.(N5)

Not giving up and overcoming barriers were normalized for some men who had survived war and displacement. The need to establish oneself and not give up speaks of these men’s resilience. However, English competency made some men vulnerable to exploitation and discrimination. This view was shared by another participant, Figure 1:

“*I looked for work a lot and tried working. Work needs strong language ability. A person is protected if they have a strong language. If one’s language is minimal, ‘half in half’ or less, then you are exposed to everything—exploitation and persecution. This is why I am doing this part-time job, because it does not require too much English … This goal for me … [Continues in Arabic] meaning this is my hope, I came to Canada with zero English … but I did not give up. I learned and reached level four. I took my citizenship. I tried to work and contribute and interact with society. I faced a lot of challenges… End of the day, I tried focusing on my children, and invest in them… I took this photo from the same area, near the home. The photo includes the moon, and I captured it behind this barrier. I consider this barrier my language barrier, and the moon my hope*”.(N1)

Most of the men were sensitive to feelings of persecution and exploitation because of war and displacement. When describing their resettlement, men spoke about ‘not giving up’ and focusing on their children’s future and having hope. This meant that for some men, fathering was about caring about their children’s future and giving up their individual pursuits and ambitions.

Most men also engaged in low-wage jobs that did not require high-level English, because they felt that they would experience greater exploitation in high-level jobs which generally required high levels of English. Other men felt discriminated against when working because of their lack of English: “So that makes you feel a bit annoyed. And why are you discriminating on me because of my accent or lack of English?” (M1).

In some cases, however, working for less money and working part-time were also perceived as protective mechanisms that made some men felt less vulnerable, Figure 2:

“*Unfortunately, now we are forced to work any job, which is causing us problem. Any job you want lets you down due to experience here in Canada. We are not getting an opportunity to work in Canada. So where do we get started if we cannot access any Canadian company? We cannot. The significant barrier is language … I applied to work [name of store]. … and I did not get recruited. … feeling down and upset. Again, with the language barrier obstructing our efforts. Why we are being tortured with the language, unable to learn, and what is the reason? … Those wood pieces in the middle of the track represent our English learning journey. We are standing on the first four or five tracks. … for us to work here like proper human beings … I still have a long steps and long way to understand the language in Canada*”.(M1)

Similarly, another Syrian man described feeling let down when his experiences were not recognized by his employer. In this context, language barriers were a structural barrier that was invisible:

“*My wife and me study hard and to gain language and a lot of things we did actually … I’m here in Canada. I’m in the … paradise, promised land. … But there are barriers, barriers, barriers. Let me explain in another way. I said it because this wire is very tiny and like, you will not recognize them until you will be closer and closer to the fence. OK? … You can’t touch everything. But when you be there, you can’t go and touch that tree, because this fence in front of you. You can’t see it*”.(M3)

Overall, most of the men felt marginalized because of their lack of English language proficiency. This process of exclusion intersected with the men’s time and stage of life.

### 3.3. Time and Stage of Life: “I Don’t Feel Secure at the End of My Life”

For many of the Syrian men, their time and stage of life posed additional challenges to obtaining meaningful employment and thus economic stability. These factors also placed additional pressures on their gender roles, Figure 3.

“*Yeah, I can’t, I don’t have a lot of time for English, and I need to work, I need to make money for my expenses, my family in Syria expenses—I have a lot of expenses. And … how many years I need to spend for change my level from example six to twelve, and we will access for ISLTS or TOEFL. I need maybe two or three years, right? … Like … don’t have money and [I] don’t have time for certificate. [I] work now with delivery … more ten thousand deliveries in Canada with Skip [the Dishes], Door Dash, and Uber Eats … I lost two cars … because the motor is broken because every day, ten-hour drive. … Now we can’t work … its very expensive … a lot of expenses for maintenance and price for delivery is very low, three or four dollars for one trip, like my hour, eight or ten dollars—I can’t continue work*”.(N2)

The same man reflected on his past as needing to work hard from a young age:

“*I can imagine, yeah, so just, yeah, so this is like people, they become older than they look…so whenever [I see] this picture [I am] reminded [of myself] … being very tired, weak, lack of sleep and, you know, you don’t have even time to eat … hands [are] very dirty, clothes … dirty. He doesn’t even have time to clean himself; he just wants to eat and have this. [I] can’t find a job, because I need to upgrade. I don’t have time, because I have other expenses … its very expensive … in Canada. I don’t know why you need to start from zero. You have to go to school to get a grade certificate—you need Canadian certificate*”.(N2)

Although holding multiple jobs and performing platform-based work allowed some men to overcome language barriers, many were cognizant of their age and stage of life. In this context, learning skills intersected with language and other literacy skills required for knowledge-based economies. For example, one man explained how the context of displacement disrupted his education and skills needed for entering the knowledge-based economy:

“*But the age plays a big challenge for us, for all of us and even for studying … like when you go to study with—like, most of the guys are like, for example, English typers long time ago. So you can tell the difference when I’m like typing slowly and the others like typing fast, or like, I like—I was the only guy with like a notebook and the pen. The others were, like, taking notes through the typing. Right?. So, it was challenging*”.(M2)

The same man went on to further explain that not having computer skills affected his ability to find work and was related to his stage of life:

“*I think the typing skills comes with the younger age … like trained better and, like, high schools, or like, especially with, like, different type of alphabet. And I’m talking about, like, totally different alphabet. … You have to think about, like, what you are typing. Are you writing, correct? You know, like. when I write, for example, the paragraph without looking at the screen, I would be like making a lot of red lines under the—under the words. … to be honest, it’s holds you back a bit … to find a job*”.(M2)

Another participant explained his perspective on how age and stage of life affected his ability to find work in his chosen profession:

“*Because here you cannot work as an architect in designing, unless you know how to deal with the computer programs I don’t have. You have to learn it from scratch to learn. I don’t have time. I’m not prepared to go through it, because I’m now 50 years old … It will take me a long time, and I am very practical. I don’t like working on computers. I like to move and be on site and make a project … planning for the project. This is what I mean most of my life*”.(M4)

Most men simply felt that they did not have time to learn the new skills required to obtain the types of jobs they wanted. This lack of time caused some men to look for alternative occupations—that is, jobs that did not require higher language ability or literacy skills. In addition, Syrian men worried about having financial security due their age and stage in life.

For example, one man described feeling excluded from being able to enjoy retirement in Canada, Figure 4:

“*I don’t see a retirement for me, I. Because starting from this age, you lost, for example, like 20 years of the pension plan or retirement plan. So, I don’t believe that … will be enough to survive, and I can tell that from seeing at work people who are 70 years old or something still working hard … when I look at those people, I feel like, ‘OK, that’s my end.’ … Those people were, like, working since the age of 20 in this country. So, they are building their retirement plan … But, I don’t feel secure at the end of my life. … So, they are using RVs as a house now, because it’s impossible to buy a house*”.(M2)

Syrian men who arrived in Canada felt increased stress related to the pressure of time and their stage in life, which posed additional challenges to obtaining economic independence. For middle-aged men, their age seemed to represent a lack of opportunities, Figure 5:

“*I recently got an old car that drove hundred and a bit. The meaning and metaphor is that days are passing by. I felt sad when it reached this number, not due to the car: we are aging, and time is passing … when we are youth, opportunities are much more, in comparison to when people are above 45*”.(N1)

Similarly, another man explained how working multiple jobs, caring for his personal needs, and finding balance affected his mental health:

“*I have to go, like, a minimum of three or four days a week, because I’m in this phase of my life is that I have to focus on my, my physical and mental health. So, the gym is not—is not something extra I’m doing. It’s essential because, after 35, your body starts to go down, and I’m almost 30, and it went so fast. I, like my entire life, …my 20 to now—till now, like this. I didn’t have the time to go to the gym. I didn’t have the time to think. I was just me going and fighting*”.(M5)

The factors of time, stage of life, and language barriers also intersected with Syrian men’s mental health and their experiences of isolation and loneliness.

### 3.4. Isolation and Loneliness: “I Feel Like I’m Alone Here…”

Some Syrian men arrived in Canada with limited family networks or social support. Single and younger men experienced increased isolation and loneliness, as a younger man explains, Figure 6:

“*Yeah … so, this photo represents what I feel now … my biggest problem here in Canada which is I feel lonely. When I was in Egypt, I had lots of friends, many more than I can count everyone … OK, so coming here and leaving all that behind, it kinda make me feel something I haven’t feel before, which is lonely. And it kinda make me afraid to talk to anyone new, because I don’t know what the results will be and being alone is … not a nice thing you know. … I used to go to cafés, to go to beaches back in Egypt with my friends. I used to have fun … now I stop all of these things. I don’t play video games; I don’t go to bar … so the things that used to be fun to me are not fun anymore*”.(N4)

It was difficult for younger men to connect with peers in Canada. Although social media and technology allows for increased connection, time differences posed additional challenges for younger men. For many men—particularly those who arrived without a social support system—socialization and community were important to their sense of belonging and identity. Culturally, some men wanted to socialize after work, as this was part of their life back in Syria:

“*The other day, I was just saddest moment in my life. I guess I was, uh, I finished work, and it was around 11:30[pm] in cold, dark, lonely … There was no one on the street. 11:30 at night. […] Yeah. Which is a festival in …in Syria. Just, like, everybody around 11:00 … do you have a coffee at 11:00? You have a falafel center or wherever you want. Everybody’s there, but here, it’s just, like, dark and empty*”.(M5)

Men who were working in platform work and low-wage jobs also experienced increased isolation and loneliness. In this context, men felt that they did not have opportunities for conversation and socialization with other employees. As one man explains, working with only Arab-speaking colleagues challenged his ability to improve his English, Figure 7:

“*…. My language capacity is not improving, because I cannot socialize with others except Arabs. Anywhere I go, I find Arabs in my face [expression] at work, which does not allow me to practice*”.(M1)

Syrian men were further excluded in job sites where they did not have opportunities to practice their English-speaking skills, but also opportunities to develop friendships and feelings of belonging. Similarly, another man explained how he felt he isolated and alone while going to work, and how it affected his mental health and wellbeing, Figure 8:

“*Waiting the bus to move to … the job because I couldn’t find anything at that time. … it started 5:00 am … I didn’t have a car. … So, I had to leave early and get two buses, so that I can arrive [on] time. … Like, even when the bus moves it, it only has three or four people on it. So, you feel like you’re there yourself, only. So, it just might reflect my feeling at that time, like, people sleeping and had to wake up early just to get there to work somewhere. Mm hmm. It was the only job I could find at that time …. They didn’t ask me for any English or anything. … They don’t care about the language; they just want some people to stand behind a machine and do some stuff*”.(N5)

Most men who were working long hours in service-industry jobs did not have opportunities to connect with other employees. Some men working in platform-based work were required to spend most of their day driving, which, in some cases, also shaped their feelings of isolation and loneliness, Figure 9:

“*When I took this picture, I was at work. I took the picture while I am inside the [name] van, as you can see from the car’s window. I was at work delivering packages. When I arrived at one destination, I found a … sunset. I love this kind of view; it’s very beautiful. You see how the sun reflects, and the sunshine has a colorful view. That makes me calm and peaceful. I took this picture and sent it to the research team. Also, … I take this picture, because sometimes, like this work at 9:00 o’clock and in this photo, it looks like it’s going to be nighttime soon. Yeah, nighttime … sometimes I work to 9:30 or 9 o’clock [pm] because we start, we start as a 11:00 o’clock [am] we start*”.(N3)

Finding moments to reflect gave some men opportunities to express positive emotions. All men expressed that having opportunities to connect with others, family, and friends provided as sense of belonging and cultural identity.

### 3.5. Belonging and Identity: “We Have Friends Gathering with Our Kids as the Kids Play with Others”

Belonging and identity were closely tied to social networks, community, kinship, and family, as one participant explained, Figure 10:

“*This picture, I took in the month of Ramadan. We break our breakfast with my friends and family. … The meeting was great and spent a nice day. I like the view that encourages me to take this picture. While we have friends gathering with our kids as the kids play with others. We spent a nice time. This kind of gathering helps me with releasing work and life stress, especially during the holy month of Ramadan*”.(N3)

Similarly, another participant explained how, for him, being a new father in Canada created a sense of belonging and identity, Figure 11:

“*When I come back from work, when it’s hard day … outside and … I put it [son’s picture] … right off the door when I open it … It really gives me give me power … [when] I got married, something became different my life. … I became a father … I always thought, ‘I don’t have family here’ … I always remember my family back home, and sometimes we sit together or have dinner or something. I have a family [now]*”.(N5)

Not all participants arrived in Canada with their families. For single men and men in relationships, having their own family in Canada created a sense of purpose. The new father quoted above explained how he was offered more pay for longer hours, but his need to work did not outweigh his need to be with his family:

“*I had to travel to [place] to do some job this job, … when I went there, they asked me to stay. They have a camp there. So, they asked me if I felt okay to stay there for a month …But I refused. I told them, ‘I have a family. I have a boy; I have to pick him up, the kids from daycare every day.’ … I can’t just go there and leave them alone here. And they are new in this country … It’s not only money, like, they pay more. …So, some people choose to go there, but in my situation, I just say ‘no.’ …. It’s not only money, …. It’s a hard heart. It’s tough*”.(N5)

Many men talked about their responsibilities for childcare, like picking children up from school. Men who had families of their own were engaged in caring responsibilities within the home, and this was part of Syrian men’s roles as fathers. Another man stated how having a family gave him new hope, Figure 12: *“This is the picture of my daughter in Canada … For us, this was, God willing—this was a glimpse of hope and light”* (M1).

Syrian men’s identities are tied to their sense of protecting and taking care of their families in and outside of Canada. This man explains how important his identity as provider is, Figure 13:

“*The first thing of my priority, I want to keep my position … to support my family. … This photo expresses that displacement of my people with one difference between the situations: … that these birds fly in the blue sky … But what about the rest of my family and friends? One of the most difficult difficulties here in Canada is that one of us [must] take care of himself, and his family here, …*”.(M3)

Although most of the men in this study felt responsible for their family’s wellbeing, having a sense of hope and family was interconnected with the Syrian men’s identities as provider, father and caretaker. For some men, identity was also related to their self-image:

“*Self-image, money isn’t it. It’s who I am. I am an artist. And even if I started doing something else, I can’t eradicate that part of my identity. Yes, I am an artist. Yeah, I’m going to always be creative. Yeah. It doesn’t matter if I’m doing it for a living or not. It has to be around*”.(M5)

Finding places to belong deeply affected Syrian men’s gender-based stress during resettlement in Canada.

### 3.6. Gender-Based Stress: “Always Under Pressure”

Overall, Syrian men experienced gendered stressors intertwined with masculinity and their identities as providers for their families, including providing remittances for families in Syria and other countries of displacement. The main source of stress was related to earning an income to support their families. These sources of stress occurred in both pre- and post-migration contexts. As one man explains, going to a neighboring country allowed him to avoid military services but did not afford him the opportunity to work in his field:

“*Actually, every year I work for three or four months in winter. There are no jobs for engineers, … especially for Syrian, because the government in Lebanon, …they didn’t allow engineers, Syrian engineers, doctors, lawyers, all of them to be in the part of their associations. They didn’t allow us … Yet we work with very cheap salaries [like the black market]*”.(M3)

Post migration and resettlement, Syrian men continue to experience lack of recognition for their experiences and skills, added to their gender-based stress. This man described feeling pressure and increasingly stressed, Figure 14:

“*In Canada, you feel stress and always under pressure … this cup of coffee, … It sort of reminds me … in Turkey, even though we lived under pressure, but we had very beautiful time and vacations. … We didn’t feel this. It’s kind of, uh, stressful … Yeah … financial stability gives everything a better taste*”.(N6)

Another single man described the effects of working multiple jobs and his mental health, Figure 15:

“*I already had, like, three jobs, so I wasn’t sleeping enough. I was overworked and I was stoned all the time, and I was going to the gym and trying to keep pushing myself forward to the point where I almost broke down. Right? That’s why I got my new job and then quit on my three other jobs. You know, no more. But that was too much; like, I was physically impacted by the depression. I’m saddened, depressed. I barely walked. I couldn’t speak*”.(M5)

Working long hours also meant that some men were not able to carry out their child-minding responsibilities or care for a spouse living with a disability, as another man explained:

“*And, and she can’t drive. She speaks good English, she has good certificate, but she can’t work. This my responsibility—I am responsible for home. … But I—it’s very hard. I tell you, it’s very hard*”.(N2)

Syrian men engaged in caring work outside of their employment. However, not having flexible hours to look after family members or pick children up from school posed additional challenges. Syrian men managed their mental health by finding time to enjoy nature, but men also described being triggered by traumatic memories, as this man explained, Figure 16:

“*So [we] went to the [NAME] Lake and this is the picture … this kind of boat, remind [me of] friends which they drowned on the Mediterranean Sea, …so most of [my] best of friends, they try to leave Turkey, going to seeking for resettlement. But, unfortunately, they pass away. So [I don’t] like to go anymore*”.(N2)

Having friends and family offered points of connection, broadening Syrian men’s sense of belonging and identity. Men found ways to belong and manage their stress by spending time with family and friends through cultural traditions and place—including spending time outside. This man explains his resilience and ways of managing isolation, Figure 17:

“*I tried to work and contribute and interact with society. I faced a lot of challenges. It is okay. End of the day, I tried focusing on my children and invest in them. How do I say—when a person reaches their maximum ability to cope with stressors! This is a coping strategy. Canada allows me to go out and change my environment to cope. If I stay at home and with the same social circle, I will fall in depression. It is a must to go on picnic and enjoy the outdoors every now and then. Canada enjoys incredible nature*”.(N1)

## 4. Discussion

This study advances research on refugee men’s experiences and provides a nuanced understanding of critical aspects of masculinities that shape refugee men’s mental health. The study draws attention to how hegemonic masculinities continue to shape Syrian men’s access to economic resources in pre- and post-settlement contexts. Five intersecting themes encompass Syrian men’s perspectives and experiences of the factors shaping their mental health in the context of their employment and resettlement in Canada: (1) language and literacy barriers, (2) time and stage of life, (3) isolation and loneliness, (4) belonging and identity, and (5) gendered stress. The findings are not mutually exclusive; rather, the themes inform each other.

Syrian men perceived language and literacy barriers as sites of social exclusion in the work force. Many men found it challenging to attend English language classes or other forms of educational programming that could advance their careers. Our finding supports previous research that suggests that refugees are disadvantaged in the labor market due to gaps in their language proficiency [69,70,71]. In addition, refugee men are less likely than women to complete Language Instruction for Newcomers (LINC) classes, a finding which may be related to the gendered pressure to be a male provider and earn an income to support their families. The men in this research reported an average of level 6 on the Canadian Language Benchmark (CLB). According to the CLB, this means that most of the men in this study had an intermediate language ability measured as an ability to function independently in daily social, educational, and work-related life [68].

However, having language skills means more than being able to read, write, and listen. Non-native English speakers may have skills at different benchmarks or stages; for example, while they might have a listening benchmark level of 6, they could simultaneously have a speaking benchmark level of 2 or lower. The average age of Syrian men in this study is 39 years; their age and time and stage of life intersecting with forced displacement and migration created additional barriers to re-learning new literacy skills required for a knowledge-based, digital economy—for example, typing in English or navigating online job sites and other forms of digital technologies. Canada is a knowledge-based economy that centers on skilled labor and education—an economy in which newcomers, including refugees, are underrepresented [69]. Research shows that the strongest predictors of labor market success in Canada are proficiency in English or French and educational attainment [72]. In our study, six out of eleven men—almost 55%—came to Canada with high levels of education, and two out of three had arrived through a private sponsorship. However, all men in this study perceived a lack of credential recognition of skills earned prior to arriving in Canada as a significant barrier.

Most men were employed in entry-level jobs or platform-based work and used their social capital—that is, their own social networks—to gain employment. Also, in this study, 73% of the men were employed in food delivery, service industries, or platform, ‘gig economy’ work, for which they did not require high levels of English language proficiency. Syrian men experienced simultaneous disadvantage and privilege where platform work provided a solution to mitigating language barriers but also disadvantaged men, as they needed to work long hours, which may take time away from their families and gendered responsibilities for child and family care. Applying intersectional analysis to hegemonic masculinity helped to expose the power relations involved in Syrian men’s marginalization built on inequalities rooted in race/ethnic, class, and gender differences. As Western economies move from labor-based economies toward knowledge- and service-based economies, these transitions may add to refugee men’s vulnerabilities, emotional isolation, and distress.

Emergent research on migrant workers and the gig economy has shown that gig platforms across the globe are increasingly dependent upon the labor of migrants [35,36,37,38]. These platforms extract maximum value from their workers, with little-to-no investment in their physical, mental, or economic security [35,36]. Platform work exists within hidden power relations that are highly racialized and exploitative and that also function to contain and manage migrants’ sense of belonging [36]. Structural conditions such as perceived discrimination, long hours, and isolating work environments—for example, delivery driving—added to the men’s sense of isolation and limited their opportunities to advance in English. Our findings align with a recent study showing that 72% of immigrants are middle-aged men who engaged in ‘gig’ or platform-work, and the majority were not recognized as having Canadian experience and gave up other career aspirations [37].

In our study, most of the men felt, as M1 expressed, “forced to work any job” to earn a living and meet the demands of providing for their families. Findings from our research also highlight a trend of economic downgrading and marginalization of Syrian refugee men in both pre- and post-resettlement contexts. The men perceived belonging through their social relations within and outside of the labor market. The implication of these factors is that the men may experience increased psychological and emotional costs beyond socioeconomic ones. Importantly, belongingness is connected to social and structural conditions [3,73]. In other words, social conditions can enable and interact with one’s individual capacity for human interaction and connection. Masculinity scholars argue that social belonging is a crucial element of class status and closely linked to distributions of power [16,19].

A study exploring the factors and processes that facilitated or hindered the experiences of belonging in Canada among 15 diverse refugees found that, amongst other factors, social connection, cultural identity, and English language proficiency were important for promoting mental health and wellbeing in the context of resettlement [73]. However, unwelcoming environments caused significant psychological distress and feelings of rejection, particularly when these environments were fostered by people in positions of power [73]. Importantly, belonging is a fundamental human need, and where social connection plays a key role for people with forced migration backgrounds [74,75]. These findings resonate with the experiences of the men in this research, who experienced isolation and loneliness in many places of employment. Isolation and loneliness were also experienced by single men who were younger and who did not have peer networks or extended family networks. Research reveals that single diverse groups of migrant men experience significant challenges in social connection inclusive of developing intimate partner relationships, due to lower social, economic, and cultural capital [76]. These findings underscore the significance of how migrant men are socially positioned through social and geographic dimensions. All the men in this research were new in their resettlement and arrived in Canada between 2016 and 2022. Evidence suggests that belonging and social connectedness through formal and informal networks can act as protective mechanisms for refugees’ mental health, particularly for those that have experienced disruption of social connections, including social life and contact with lifelong friends and relatives [75,76].

For Syrian men who had families, their identity and sense of belonging were strongly connected to being a provider of their family’s material and financial needs. Some men in our study took on intensive caring roles for their children. Investing in their children’s future and having family connections and support were sources of prosperity and hope for the Syrian men in this study. This is consistent with a study conducted with Syrian refugee fathers in Canada which found that Syrian fathers experienced changes to their gender roles—changes that were not only associated with cultural and religious beliefs, but also intertwined with masculinity and broad structural conditions of being able to fulfill their role as breadwinner [41]. In our research, Syrian men with families—and those who became new fathers in Canada—experienced their families as a source of strength and social support. Paradoxically, men with families also experienced gender-based distress in not being able to financially support themselves or their families. Managing multiple life transitions including fatherhood could lead to increased emotional distress for refugee men.

The Syrian men in this study were diverse and non-homogenous. Time and stage of life impact the way Syrian men perceived their opportunities for advancement. Younger men experienced stressors related to finding work to support their families while perusing their education while in Canada. These differences—related to stage of life and time pressures—are supported in research on Syrian refugee youth in Canada, which found that refugee youth often face cultural and linguistic barriers to navigating pathways such as pursuing high education and searching for a job or housing while financially supporting their families [77]. Refugee men and boys may experience differential contextualized gender-based stress based on their age and stage of life.

Gender-based stress intersected with men’s gender roles and the feeling that they always felt under pressure. Findings from other research show that Syrian migrant women may have more opportunities than men to create community, access social and legal support, and develop their capacity for building entrepreneurial opportunities in different geographies of resettlement [12,22]. Syrian men continue to experience barriers to prosperity and economic integration, which may be related to hegemonic masculinity. This suggests that hegemonic masculinity reinforces women as vulnerable and thereby obfuscates the social and structural vulnerability of refugee men and the broad social, political, and economic factors that shape their access to economic independence. 

Mental health stigma and normative masculinity can be barriers to talking with men about their mental health [8,16,44,45]. Moreover, mental health constructs are not universal; for men with Arabic backgrounds, having poor mental health is highly stigmatized and may be thought of as a mental disability rather than the result of social determinants of health [8,10,32,40]. Photovoice provided a way for Syrian men to talk about mental health and the factors that shape their settlement and employment experiences in Canada. A recent study exploring masculinities and culture and men’s health in Australia supported a gendered relations approach to engaging men in mental health promotion work [16]. This includes providing opportunities for social connection and community-based men’s health that provide peer mentorship. This can allow men opportunities for social connection through cultural and group activities [16]. These approaches may resonate with migrant men who come from collectivist societies where normative masculinity prevents men from seeking help.

Similarly, research on Syrian refugees in Canada found predictors of mental health that included having postsecondary education, sufficient finances, proficiency in English, and control over one’s circumstances; being employed and married; and reporting satisfaction with one’s housing and low stress levels [78]. In addition, Syrian women were found to have poorer mental health and less access to resources than men. Our findings suggest that, for men, lack of employment and gender-based stress were underscored by the proficiency in English and being meaningfully employed. However, differences in reporting of mental health between women and men, may be related to the fact men are less likely to report mental health challenges. Masculinity and broad cultural discourses affect the ways in which the vulnerability of men and emotional distress are understood and reported. Our findings underscore the importance of an intersectional analysis, which understands gender roles as not fixed and influenced by multiple intersecting factors.

Some Syrian men in our research also described setting aside their own career aspirations to focus on supporting their children; other men stated that they valued time with their family over working as many hours as possible. Men also talked about needing to support their wives by driving them to medical appointments because they had physical disabilities. These findings support calls to disrupt anti-immigrant sentiments which homogenize and portray Arab men as violent and which reinforce hegemonic masculinity [14,18]. Research on refugee men who experienced forced migration from Sudan has shown that gender roles, parenting practices, and social relations are embedded within broad socio-political processes that reinforce essentialist and monolithic notions of masculinity. In contrast, Sudanese refugee men’s experiences and identities are fluid and dynamic; in between, men “wal[k] the line” between cultural and institutional gendered expectations [79]. Similarly, research on caring masculinity suggests that working-class men—including men marginalized due to class, race, ethnicity, or nationality status—embrace values of care through their relationships and not through gaining power [7,19,20]. Caring masculinity and exploration of gender relations provides an alternative model for re-imagining and understanding how Syrian refugee men perform masculinity within their cultural role as the provider, as they re-negotiate their identities during resettlement.

The Syrian men’s narratives in this study suggest that men experience multiple intersecting processes of hegemonic marginalization based on pre- and post-migration policies and practices that exclude them. The study findings also suggest that Syrian men resist forms of hegemonic masculinity and engage in caring masculinities such as fathering and supporting their children and contributing to service industries. This exposes the non-monolithic character of masculinities and suggests that migrant men need to belong through inclusive policies and practices that recognize their skills and promote their mental health.

### 4.1. Strengths, Limitations, and Future Directions

This study provides an intersectional analysis of Syrian Canadian men’s experiences, on the factors shaping their mental health and resettlement and integration in Canadian society. Syrian men’s voices provide understanding about their collective subjectivities within wide power relations and social structures of resettlement and shed light on the social determinants of refugee men’s mental health. This study adds to broad discourses that seek to disrupt essentialist notions of masculinity and to promote an understanding of men’s vulnerabilities. Drawing from the study findings, an ethos of caring masculinity may offer alternatives and resistance to hegemonic masculinities that devalue and marginalize migrant men. Caring masculinity may also offer a framework from which to promote refugee men’s mental health.

Due to the small sample size and qualitative approach, generalizations about the social determinants of Syrian refugee men’s mental health cannot be made. However, findings may have transferability to other men with experience forced migration backgrounds and similar contexts. To promote rigor, the study included iterative points of validation—community advisory board meetings, photovoice workshops, and extensive photo elicitation interviews—which support informational power, or the degree to which the information was theoretically rich and able to answer the research questions [49].

### 4.2. Implications for Research

Migration policies and patterns have shifted toward a masculinized labor market gap, meaning that more men are now filling labor shortages globally [1]. Little attention has been paid to migrant men and boys, masculinity, and the factors that support their belonging and mental health. Future research and practice must consider gendered vulnerabilities across genders, because migrants, including refugees and their families, experience far-reaching gendered impacts on their mental health. Migrant men are particularly vulnerable and at risk of poor mental health because they have a high likelihood of being unemployed or underemployed. Additional migration factors and stage of life transitions may further decrease refugee men’s resilience and add to their experiences of gender-based stress. Research on men’s mental health needs to include experience of racialized and migrant men to promote their mental health. 

### 4.3. Implications for Policy and Practice

Because hegemonic masculinity may overshadow men’s help-seeking behavior, many men may not access mainstream community resources or support for their mental health. Findings from this research suggest that peer mentorship and peer-based programs for migrant men may offer opportunities for connection, social support, and belonging. Importantly, not all migrant men are homogenous; gender-responsive programs should consider factors such as stage of life and age to provide support and mentorship intergenerationally. Although cultural constructions of mental health and illness vary across cultures, Syrian men benefited from discussing their experiences in contexts where trust was developed with peers. This finding supports previous research that recommends programing specific to men that is peer-led and that promotes men’s resilience and psychological wellbeing [16]. Employment and settlement support services could also offer peer-based support and mentorship services that promote belonging and inclusive work environments. Strengthening integration between the health sector and settlement service sector may facilitate comprehensive supports for migrant men and their families and more inclusive migrant healthcare [80].

## 5. Conclusions

In this study, Syrian men with displaced refugee backgrounds provided their voices on the intersecting social determinants that shaped their mental health in a resettlement context. The factors that affected their mental health included language and literacy barriers, a lack of time, stage of life, feelings of belonging, identity, and gender-based stress. The importance of meaningful employment to men’s mental health is underscored by structural processes that marginalize refugee men. Syrian men are particularly vulnerable to gender-based stress in resettlement contexts. Adopting an ethos of caring masculinity may disrupt harmful forms of masculinity, promote men’s mental health, and make space for migrant men to belong. Public health policies and practices need to adopt gender-responsive policies and programs that include intersectional dimensions of gender and peer-based support which promotes the mental health and wellbeing of migrant men and prevents root causes of gender inequalities.

## Figures and Tables

**Figure 1 ijerph-21-01600-f001:**
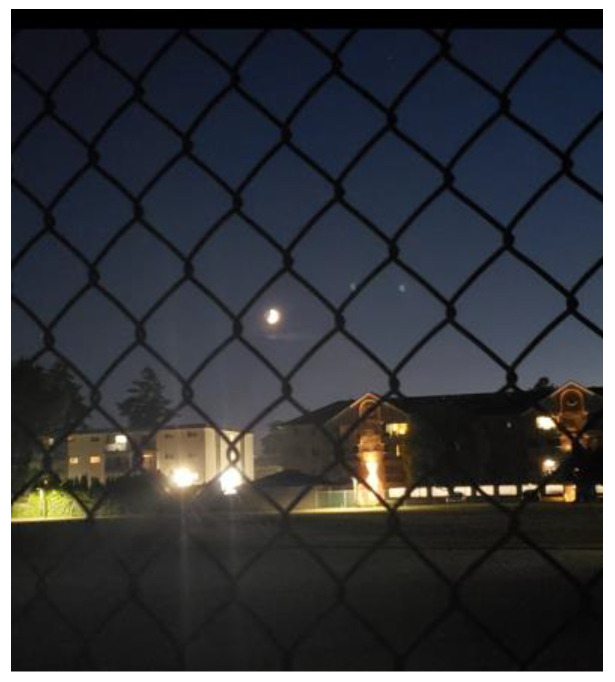
Language and literacy barriers: photo by N1.

**Figure 2 ijerph-21-01600-f002:**
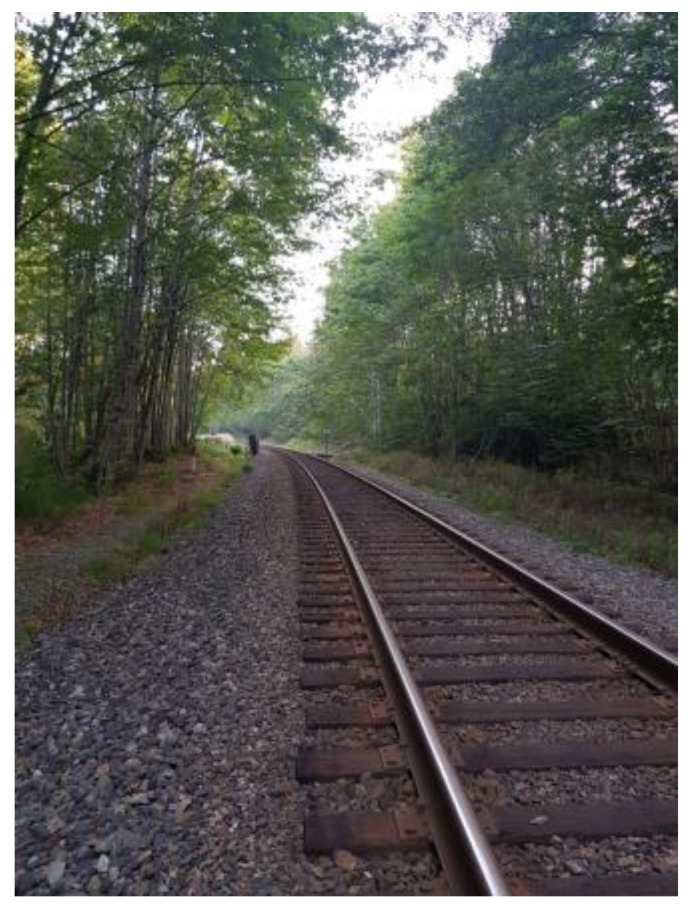
Language and literacy barriers: photo by M1.

**Figure 3 ijerph-21-01600-f003:**
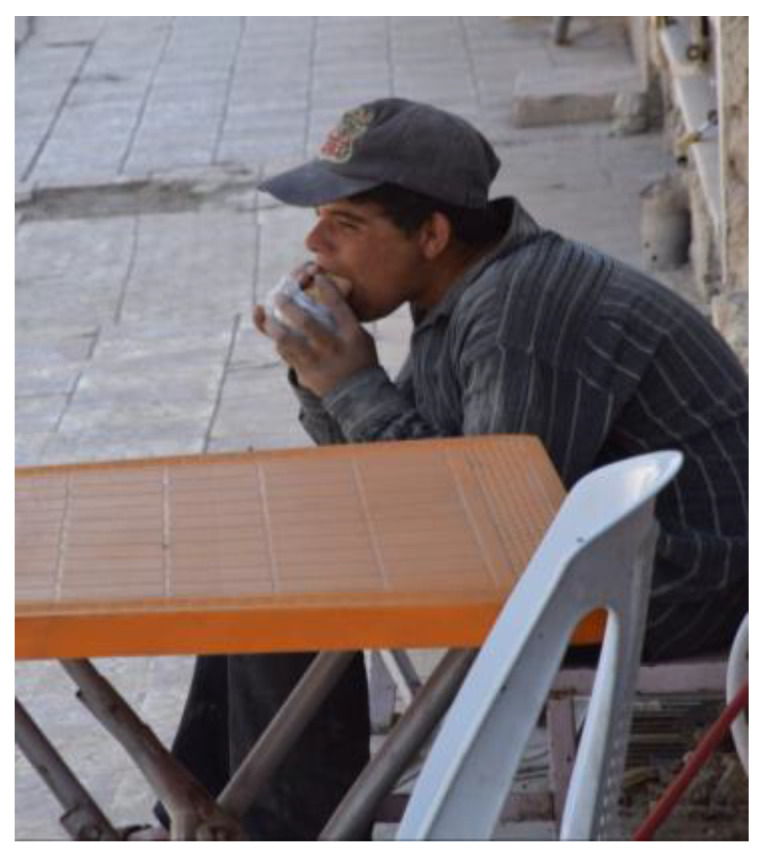
Time and stage of life: photo by N2.

**Figure 4 ijerph-21-01600-f004:**
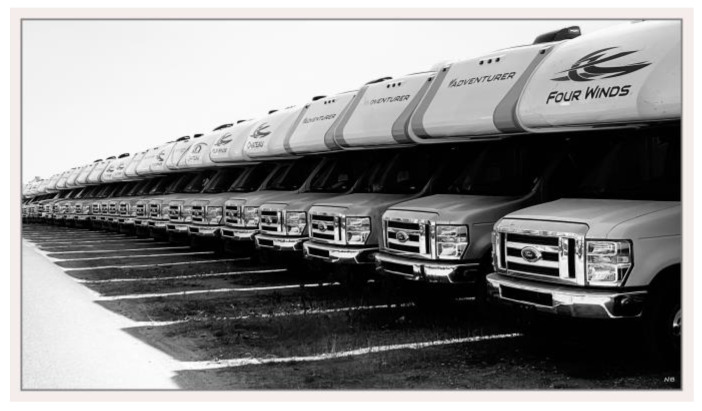
Time and stage of life: photo by M2.

**Figure 5 ijerph-21-01600-f005:**
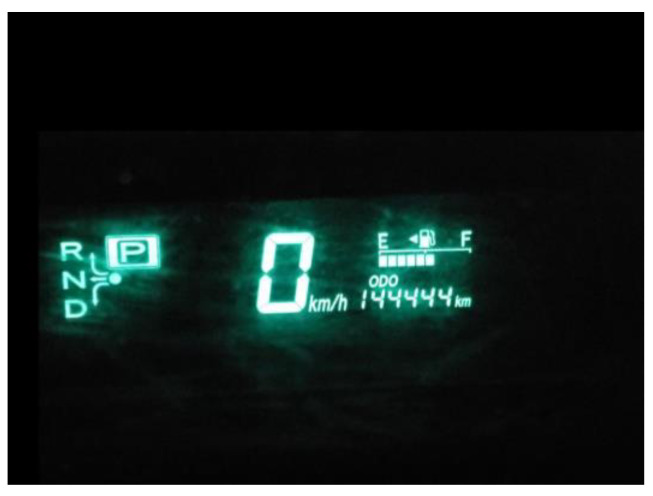
Time and stage of life: photo by N1.

**Figure 6 ijerph-21-01600-f006:**
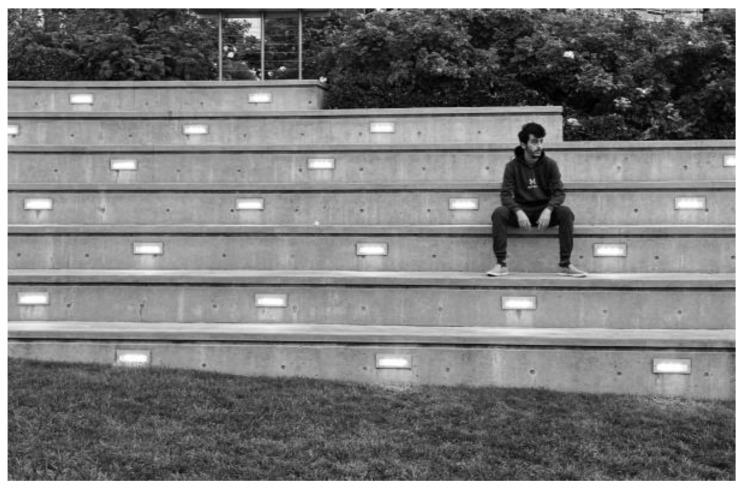
Isolation and loneliness: photo by N4.

**Figure 7 ijerph-21-01600-f007:**
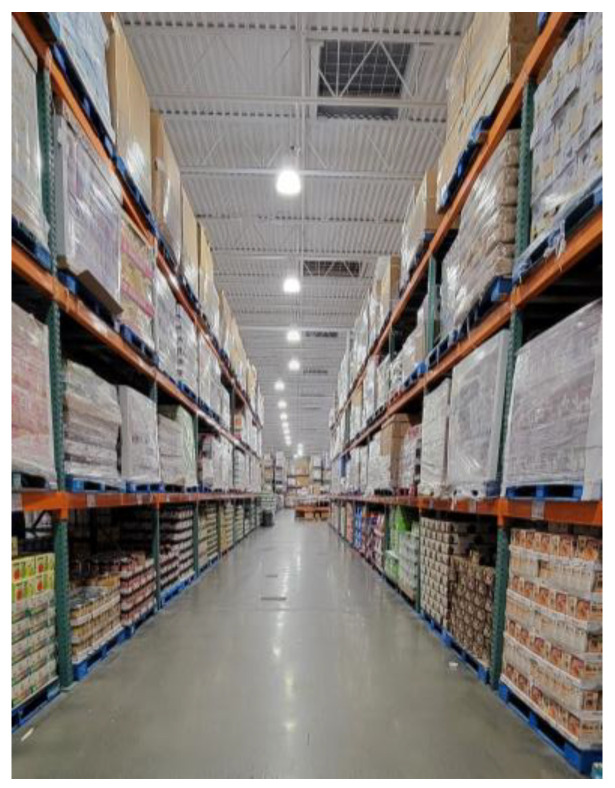
Isolation and loneliness: photo by M1.

**Figure 8 ijerph-21-01600-f008:**
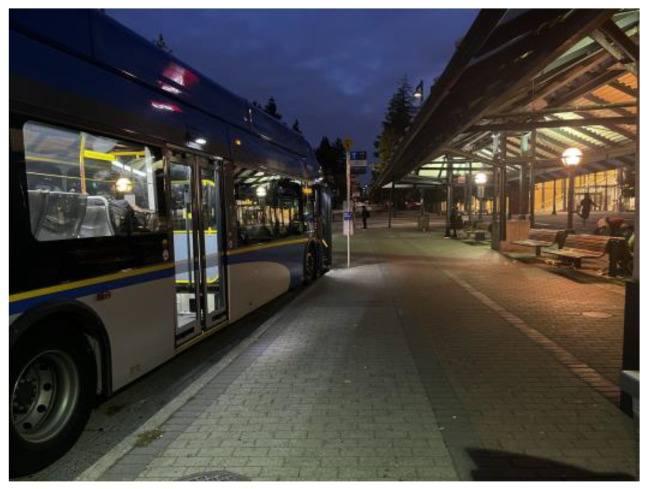
Isolation and loneliness: photo by N5.

**Figure 9 ijerph-21-01600-f009:**
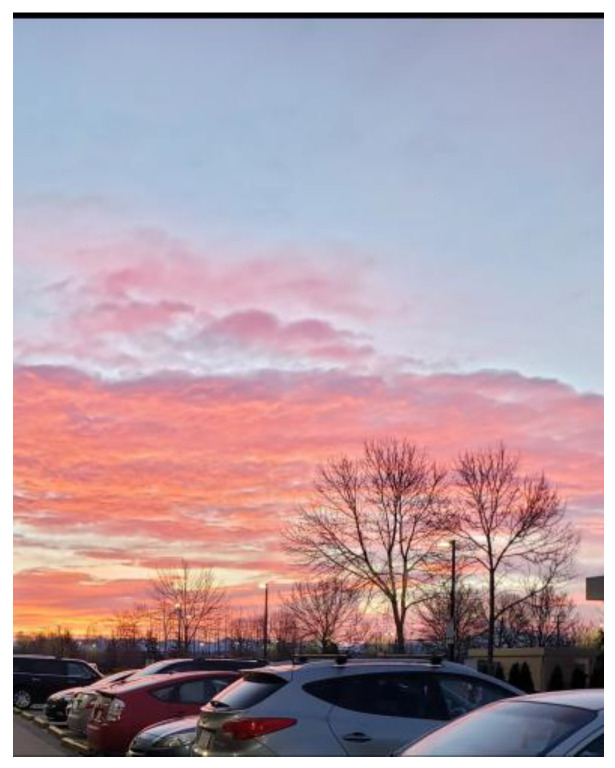
Isolation and loneliness: photo by N3.

**Figure 10 ijerph-21-01600-f010:**
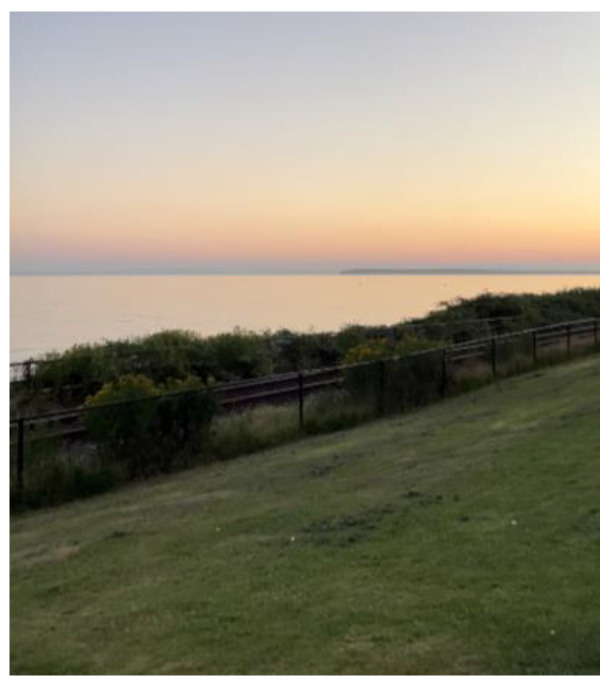
Belonging and identity: photo by N3.

**Figure 11 ijerph-21-01600-f011:**
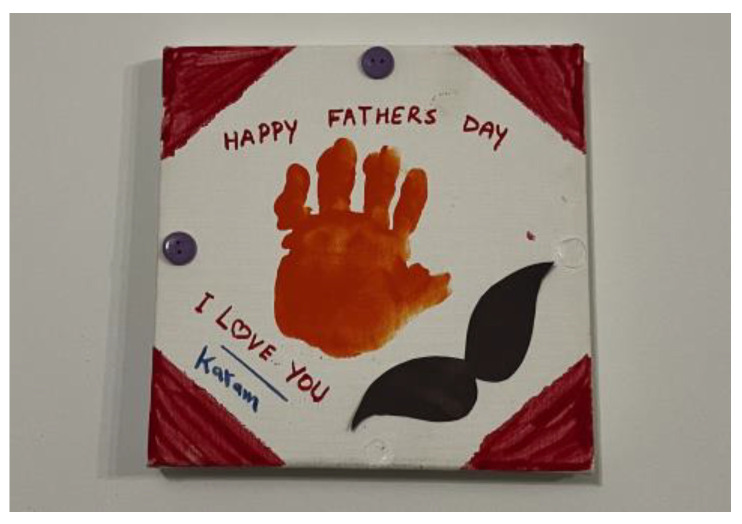
Belonging and identity: photo by N5.

**Figure 12 ijerph-21-01600-f012:**
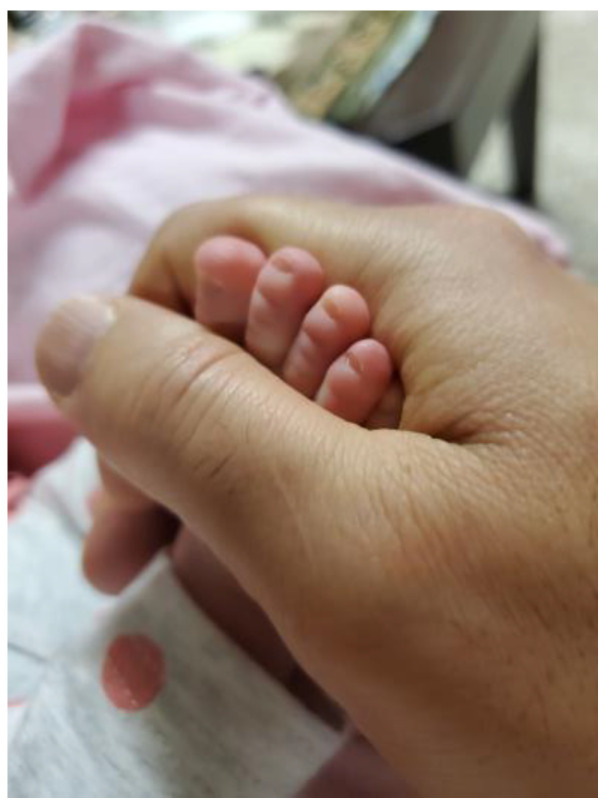
Belonging and identity: photo by M1.

**Figure 13 ijerph-21-01600-f013:**
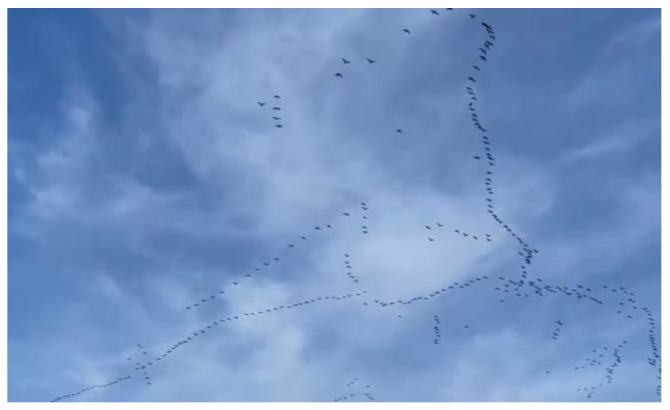
Belonging and identity: photo by M3.

**Figure 14 ijerph-21-01600-f014:**
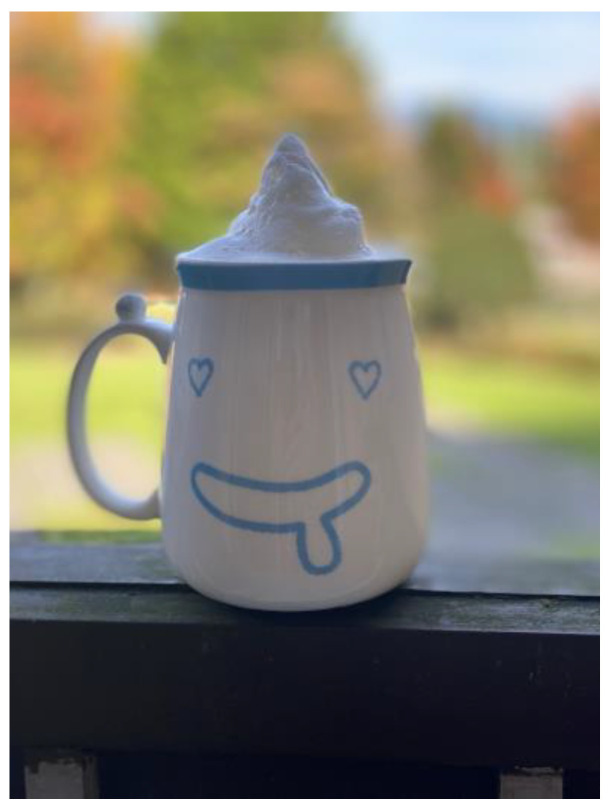
Gender-based stress: photo by N6.

**Figure 15 ijerph-21-01600-f015:**
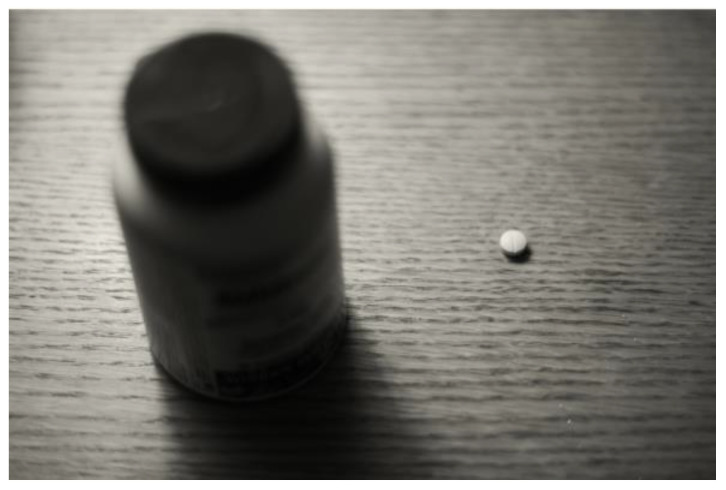
Gender-based stress: photo by M5.

**Figure 16 ijerph-21-01600-f016:**
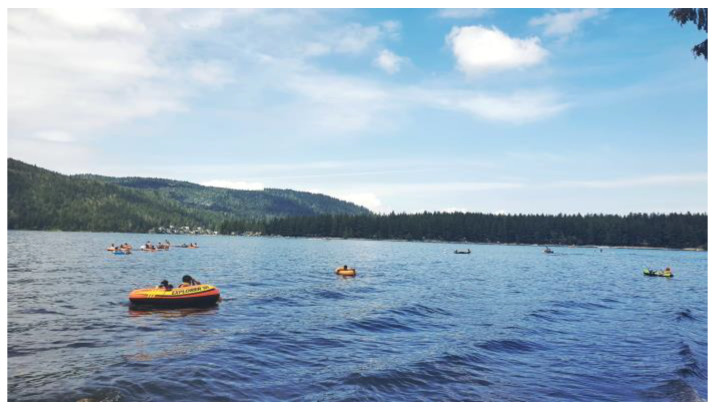
Gender-based stress: photo by N2.

**Figure 17 ijerph-21-01600-f017:**
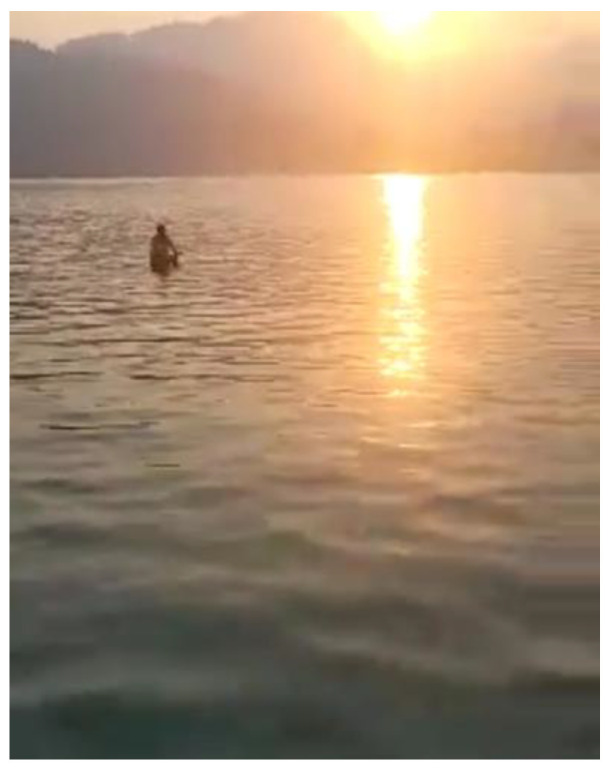
Gender-based stress: photo by N1.

**Table 1 ijerph-21-01600-t001:** Participant demographics.

Demographics		
Age	Median = 39.1	Range (25–51)
	Standard Deviation = 8.6	
Housing	Rent (n = 11)	Rent (100%)
	Own (n = 0)	Own (0%)
Type of Housing	Market Housing (n = 5)	Market Housing (45%)
	Subsidized Housing (n = 6)	Subsidized Housing (55%)
Average Cost of Rent	Average = CAD 1255	Range (CAD 800–CAD 2200)
Ethnicity	Syrian (n = 10)	Syrian (91%)
	Palestinian (n = 1)	Palestinian (9%)
Education	Master’s Degree (n = 1)	Master’s Degree (9%)
	Bachelor’s Degree (n = 6)	Bachelor’s Degree (55%)
	High School Diploma (n = 3)	High School Diploma (27%)
	Less than High School (n = 1)	Less than High School (9%)
Marital Status	Married (n = 9)	Married (82%)
	Single (n = 2)	Single (18%)
Employment Status	Employed (n = 8)	Employed (73%)
	Self-Employed (n = 1)	Self-Employed (9%)
	Unemployed (n = 2)	Unemployed (18%)
Religion	Muslim (n = 7)	Muslim (64%)
	Christian (n = 2)	Christian (18%)
	None (n = 2)	None (18%)
Country of Origin	Syria (n = 10)	Syria (91%)
	Kuwait (n = 1)	Kuwait (9%)
Citizenship	Canadian (n = 6)	Canadian (55%)
	Permanent Resident (n = 5)	Permanent Resident (45%)
Immigration	Government-Assisted Refugee (n = 6)	Government-Assisted Refugee (55%)
	Privately Sponsored Refugee (n = 3)	Privately Sponsored Refugee (27%)
	Asylum (n = 2)	Asylum (18%)
Year Arrived in Canada	2016 (*n* = 3) 2017 (*n* = 1) 2018 (*n* = 3) 2019 (*n* = 2) 2021 (*n* = 1) 2022 (*n* = 1)	Average = 2019 Range (2016–2022)
Canadian Language Benchmark (CLB)	Level 4 (*n* = 3) Level 6 (*n* = 1) Level 7 (*n* = 1) Level 8 (*n* = 6)	Average = Level 6 Range (Level 4–8)
Average Income	CAD 42,000	Range (CAD 18,000–80,000)
Average Family Composition	3.3 persons	Range (1–6 people)
Number of Children	Average 1.4 children	Range (0–4)

## Data Availability

The data presented in this study are available on request from the corresponding author and stored on the NVivo 12 software data management program, having been de-identified using codes. Participants’ photographs are available upon request as part of the data set.

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
