# Peer review of "Resettlement, Employment, and Mental Health Among Syrian Refugee Men in Canada: An Intersectional Study Using Photovoice"

_ijerph, 2024, doi:10.3390/ijerph21121600_

Round 1
Reviewer 1 Report
Comments and Suggestions for Authors
Thank you so much for submitting mental health issues among refugee population which is one of the key public health issues in the IJERPH which is significant in global health literature. Here is my feedback to improve your manuscript before accepting it for publication.
Abstract:
· Methods section needs bit clear about the data collection and data analysis which is not clear yet.
· Findings not clear what are the key findings of this paper which are not clearly articulated. There might have several intersecting themes that should be succinctly summarised and make consistent under in the entire writing. Only one sentence findings do not make this section clear. For example, you have concluded that low wage job as key issue but that should be first come in your findings then you can draw succinct conclusion and provide a pragmatic suggestion to over come the gaps.
Methods:
· The methods section is not very clear about the research design which should be succinctly stated at the beginning and followed the study sites, participants and rationale of the selection.
· What was the criteria of recruited 11 participants and how can you ensure that there is no any selection and recruitment bias.
Findings:
· Very interesting and details finding section but I would suggest to authors do you think all 23 photos which can choose most representative voices and the images that provide messages how many photos are ideal for one paper.
Discussion:
· Discussion would be much clear if authors outline at the beginning what aspect you are going to discuss under this section, Based on the authors outline, discussion should be more focused and critically articulated that makes section much more clear. I could not see any added value of line 764-773 so make this very clear first and critically articulated the section.
Author Response
Dear Reviewer 1, thank you for your valuable recommendations for strengthening this manuscript. Given the extensive revisions required we have re written and organized the manuscript, including shortening the introduction. Because it would be difficult to read the track changes version we provide a "clean" copy for your review. Please find below your recommendations and our responses for how we have improved the manuscript.
Methods section needs bit clear about the data collection and data analysis which is not clear yet.
- Findings not clear what are the key findings of this paper which are not clearly articulated. There might have several intersecting themes that should be succinctly summarised and make consistent under in the entire writing. Only one sentence findings do not make this section clear. For example, you have concluded that low wage job as key issue but that should be first come in your findings then you can draw succinct conclusion and provide a pragmatic suggestion to over come the gaps.
Thank you for this recommendation, because we used intersectionality as an analytic tool in combination with masculinities framework, we believe that low wage job is only one aspect of the overall findings. For example, we draw attention to the construction of masculinity in working class settings among racialized Syrian men. These points have also been raised by reviewer 2. Therefore, to address your comments we have woven in more succinctly the findings with the methodology. Please note that we believe our findings provide a nuanced understanding about the social structures that reinforce marginalized masculinity through multiple overlapping themes, e.g. language and literacy barriers, time and stage of life, isolation and loneliness, belonging and identity and gender-based stress. We agree on the need to provide pragmatic suggestions for public health policy and practice and have added those sections into the discussion.
Methods:
- The methods section is not very clear about the research design which should be succinctly stated at the beginning and followed the study sites, participants and rationale of the selection. Thank you we have revised the methods section to this criterion in order to provide added clarity.
- What was the criteria of recruited 11 participants and how can you ensure that there is no any selection and recruitment bias. Participants self selected to participate in the study. The criteria for inclusion were used to answer the main research question. A purposeful sampling method was used to meet the inclusion criteria of the Syrian men. In this research inclusion criteria included, being Syrian man, new to Canada (5 years) and being over 18, working, or looking for a job and have received support from a settlement agency.
Findings:
- Very interesting and details finding section but I would suggest to authors do you think all 23 photos which can choose most representative voices and the images that provide messages how many photos are ideal for one paper.
We appreciate this comment, it is very difficult to limit the photographs. There is a need to facilitate the “voice” of the participants through the photos and the varying ways that photos represent the themes and key findings. The photos correspond to the men’s individual as well as collective stories. As much as possible we want to decolonise traditional approaches to research with refugees. Using arts-based methods and photovoice is one approach to advance this methodology. However, we have removed some photographs for improved readership. We have also moved participant data to findings section.
Discussion:
- Discussion would be much clear if authors outline at the beginning what aspect you are going to discuss under this section, Based on the authors outline, discussion should be more focused and critically articulated that makes section much more clear. I could not see any added value of line 764-773 so make this very clear first and critically articulated the section.
- Thank you for this feedback. We have restructured the entire discussion to provide clarity and articulation of how we applied the theoretical frameworks of intersectionality and masculinity to understand Syrian men’s perspectives and experiences. We have also added implications for research, policy and practice section for greater practical recommendations based on this research.

Reviewer 2 Report
Comments and Suggestions for Authors
I read the article “Resettlement, Employment, and Mental Health Among Syrian Refugee Men in Canada: An Intersectional Study using Photovoice” with interest. It explores the complex experiences of Syrian refugee men in Canada, examining how employment, identity, and social factors impact mental health. This study applies an intersectional lens, along with Photovoice, a community-based participatory method that allows participants to tell in-depth, visual stories.
The article generally has a coherent flow, with well-organised sections that guide the reader through the research objectives, methodology and findings. However, certain transitions, particularly between methodology and findings, could be smoother, as the findings could benefit from a more structured link to the theoretical framework of intersectionality.
The article would benefit from a more in-depth engagement with the literature specifically on masculinities and the mental health of refugee men. The addition of references to the challenges of resettlement and the gendered stress associated with the refugee experience in Western contexts would strengthen the argument and contextual grounding.
The use of an intersectional approach is particularly timely as it emphasises the multi-layered struggles of these men in adapting to a new socio-economic environment.
The role of masculinity as a cultural and psychological construct could be explored more thoroughly as this would deepen the analysis of gender stress and identity issues. While the photovoice method is well applied and provides rich data, the theoretical discussion of how intersectionality feeds into data analysis could be expanded to clarify how specific intersections are prioritised or weighted.
At times, the article’s main arguments about masculinity and mental health seem underdeveloped. For example, when discussing gendered stress, the authors could elaborate on how masculinity intersects with cultural expectations and occupational challenges and affects mental health in unique ways.
Some sections, particularly in the methodology, are overly detailed, which can distract from the main arguments. For example, the explanation of all the procedural steps in the photovoice process could be shortened to focus more on the thematic findings.
This article is a valuable contribution to the literature on migration, mental health and masculinity. With revisions, particularly in terms of the coherence of the arguments, theoretical depth and contemporary relevance, the article would be suitable for publication in the International Journal of Environmental Research and Public Health.
Author Response
Dear Reviewer 2, thank you for your time in reading the manuscript and your valuable comments and recommendations. We have made every effort to address your recommendations. In particular we have re written the entire manuscript in order to provide more depth and clarity on men and masculinities. These arguements have been threaded throughout the new version of the manuscript. Due to the extensive major revisions required we have provided a "clean" copy of the manuscript. Please find your recommendations and our attention to them in the responses below.
I read the article “Resettlement, Employment, and Mental Health Among Syrian Refugee Men in Canada: An Intersectional Study using Photovoice” with interest. It explores the complex experiences of Syrian refugee men in Canada, examining how employment, identity, and social factors impact mental health. This study applies an intersectional lens, along with Photovoice, a community-based participatory method that allows participants to tell in-depth, visual stories.
The article generally has a coherent flow, with well-organised sections that guide the reader through the research objectives, methodology and findings. However, certain transitions, particularly between methodology and findings, could be smoother, as the findings could benefit from a more structured link to the theoretical framework of intersectionality.
- Thank you for pointing this out, we have provided discussion of link between intersectionality and masculinity to provide a better link between methodology and findings. In this manuscript we are not able to take a deep dive into all the forms of masculinities that exist, as this is the main findings paper of the research. However, there is a tension between what is known as hegemonic masculinity and marginalized masculinity and their variants. We offer the framework of these two theoretical approaches to provide a nuanced understanding of Syrian men’s social positioning in resettlement and pre migration, and the structural inequities and inequalities they experienced in the spaces in which masculinities operate. To this end we have foregrounded these frameworks in the introduction, methods and discussion sections.
The article would benefit from a more in-depth engagement with the literature specifically on masculinities and the mental health of refugee men. The addition of references to the challenges of resettlement and the gendered stress associated with the refugee experience in Western contexts would strengthen the argument and contextual grounding.
- Thank you again for making this point clear. We aim to write another manuscript advancing analytic understandings of masculinity and specifically caring masculinity as one alternative framework. However, as this is not entirely a methodological paper, we are focusing on the main findings to advance policy, practice and community based pathways for health promotion of migrant med. Something that was not made clear in the original draft. Given your points to advance and integrate literature on masculinities we have made attempts in both the introduction, methods and discussion sections,e.g. we have threaded maculinity theory throughout the revised version of the manuscript.
The use of an intersectional approach is particularly timely as it emphasises the multi-layered struggles of these men in adapting to a new socio-economic environment.
The role of masculinity as a cultural and psychological construct could be explored more thoroughly as this would deepen the analysis of gender stress and identity issues. While the photovoice method is well applied and provides rich data, the theoretical discussion of how intersectionality feeds into data analysis could be expanded to clarify how specific intersections are prioritised or weighted. We agree with your comments and have made attempts to integrate masculinity as a central construct within the manuscript.
- Thank you for this comment, we believe the revised version of the manuscript has addressed the concept and theories of masculinity and in the methods section have described the value of adding intersectionality with hegemonic masculinity with added references.
At times, the article’s main arguments about masculinity and mental health seem underdeveloped. For example, when discussing gendered stress, the authors could elaborate on how masculinity intersects with cultural expectations and occupational challenges and affects mental health in unique ways.
- Thank you for this recommendation, we have aimed to strengthen the article by threading theories of masculinity throughout the manuscript. To add this theory to findings section only may not be as concise so we have brought attention to masculinity theory and men’s masculinities throughout the article. We have also added key references on masculinities and migrant masculinities to the article.
Some sections, particularly in the methodology, are overly detailed, which can distract from the main arguments. For example, the explanation of all the procedural steps in the photovoice process could be shortened to focus more on the thematic findings.
- We have paid attention to detail in this section because the methods of photo voice combined with intersectionality are complex in and of themselves. However, we have shortened and edited this section for clarity and included a note on rigor. Also to promote rigor of the qualitative study we provide details to enhance auditability and validation of the findings. However, in order to address your recommendations we have edited the methodology section to provide clarity and removed repetitive statements.
This article is a valuable contribution to the literature on migration, mental health and masculinity. With revisions, particularly in terms of the coherence of the arguments, theoretical depth and contemporary relevance, the article would be suitable for publication in the International Journal of Environmental Research and Public Health.
- Thank you for your valuable comments and insights. We hope the revisions add theoretical depth while providing practical guidance and recommendations for public health actions for transforming gender responsive care for migrant men in resettlement contexts. Introduction includes definition of masculinity and provides a clearer orientation to the study and manuscript. We hope that this strengthens the article and is more engaging for readers.
